# Exercise and Curcumin in Combination Improves Cognitive Function and Attenuates ER Stress in Diabetic Rats

**DOI:** 10.3390/nu12051309

**Published:** 2020-05-04

**Authors:** Jin Ah Cho, Se Hwan Park, Jinkyung Cho, Jong-Oh Kim, Jin Hwan Yoon, Eunmi Park

**Affiliations:** 1Department of Food and Nutrition, Chungnam National University, Daejeon 305-764, Korea; jacho@cnu.ac.kr; 2Department of Sport Science, Hannam University, Daejeon 34430, Korea; psh8179@gmail.com (S.H.P.); jokim1019@hanmail.net (J.-O.K.); 3Institute of Sports & Arts Convergence, Inha University, Incheon 22212, Korea; lovebuffalo@gmail.com; 4Department of Food and Nutrition, Hannam University, Daejeon 34054, Korea

**Keywords:** curcumin, exercise, diabetes, cognitive function, endoplasmic reticulum (ER) stress

## Abstract

Type 2 diabetes mellitus (T2DM) is a metabolic disease associated with chronic low-grade inflammation that is mainly associated with lifestyles. Exercise and healthy diet are known to be beneficial for adults with T2DM in terms of maintaining blood glucose control and overall health. We investigated whether a combination of exercise and curcumin supplementation ameliorates diabetes-related cognitive distress by regulating inflammatory response and endoplasmic reticulum (ER) stress. This study was performed using male Otsuka Long-Evans Tokushima Fatty (OLETF) rats (a spontaneous diabetes Type 2 model) and Long-Evans Tokushima Otsuka (LETO) rats (LETO controls) by providing them with exercise alone or exercise and curcumin in combination. OLETF rats were fed either a diet of chow (as OLETF controls) or a diet of chow containing curcumin (5 g/kg diet) for five weeks. OLETF rats exercised with curcumin supplementation exhibited weight loss and improved glucose homeostasis and lipid profiles as compared with OLETF controls or exercised OLETF rats. Next, we examined cognitive functions using a Morris water maze test. Exercise plus curcumin improved escape latency and memory retention compared to OLETF controls. Furthermore, OLETF rats exercised and fed curcumin had lower IL6, TNFα, and IL10 levels (indicators of inflammatory response) and lower levels of ER stress markers (BiP and CHOP) in the intestine than OLETF controls. These observations suggest exercise plus curcumin may offer a means of treating diabetes-related cognitive dysfunction.

## 1. Introduction

Diabetes is a disease, in which dysregulation of insulin signaling leads to insulin resistance [1]. Clinical studies have shown that glycemic regulation and disrupted insulin singling negatively affect brain function [2]. Furthermore, several studies have shown diabetes is a risk factor of cognitive-related diseases such as vascular dementia and Alzheimer’s disease [3]. Accordingly, elderly patients with Type 2 diabetes (T2DM) are given clinical advice on the beneficial effect of exercise to increase insulin sensitivity [4]. However, few comprehensive guidelines have demonstrated the relationship between cognitive dysfunction and diabetes.

Endoplasmic reticulum (ER) plays a major role in protein synthesis and in folding and processing of secretory and transmembrane proteins [5]. When misfolded or unfolded proteins accumulate in ER due to genetic or environmental factors, ER stress occurs, which is mediated by three ER transmembrane proteins, IRE1, PERK, and ATF6, inducing unfolded protein response (UPR) [6].

Since ER stress can activate cellular inflammatory pathways that impair cellular functions and lead to metabolic disorders, existing evidences suggest that ER stress is a key link between T2DM, obesity and insulin resistance, [7,8]. It is known that exposure to high glucose levels can induce ER stress via the generation of free radicals, aberrant protein glycosylation, and increased membrane and protein turnover, leading upregulation of C/EBP homology protein (CHOP) expression (a prominent mediator of ER stress-induced apoptosis).

Diabetes is also associated with cerebral atrophy, neurological disabilities, cognitive decline, and psychomotor dysfunction, called diabetic encephalopathy (DEP) [9]. A growing body of evidence indicates that oxidative stress, chronic inflammation, mitochondrial dysfunction, impaired insulin signaling, and ER stress contribute to the development of DEP. However, few studies have investigated the effect of intestinal ER stress.

It has been reported that at the beginning of ER stress, increased GRP78 expression in the rat brain prevented CHOP-induced apoptosis, but when ER stress was persistent, CHOP levels were markedly increased in the hippocampus of diabetic rats [10]. This suggests that CHOP-ER stress-mediated apoptosis may be involved in hyperglycemia-induced hippocampal synapses and neuronal impairment, and thus, promote diabetic cognitive impairment.

Curcumin is a natural compound with known antioxidant and anti-inflammatory properties [11,12]. Although it has been shown to induce apoptosis of cancer cells, its effects on inflammatory response in T2DM is not well investigated. Accumulating evidence indicates low-grade inflammation, oxidative stress, and ER stress contribute insulin resistance in T2DM. Therefore, it would appear that suppression of ER stress and inflammation might improve cognitive function in T2DM [13]. However, it has not been established whether exercise plus curcumin supplementation could improve diabetes-associated cognitive dysfunction by regulating ER stress response.

In the present study, we investigated whether cognitive functions differ in Otsuka Long-Evans Tokushima Fatty (OLETF) rat (the rat of diabetes), and in Long-Evans Tokushima Otsuka (LETO; the rat of control), and whether a combination of exercise and curcumin supplementation would ameliorate diabetes-associated cognitive dysfunction. Furthermore, we measured changes in the levels of inflammatory response cytokines and CHOP induced by a combination of exercise and curcumin supplementation in the rat model of spontaneous diabetes.

## 2. Materials and Methods

### 2.1. Animal and Experimental Design

LETO rats were used as controls (male LETO controls; *n* = 10), and OLETF rats, which a hyperphagic phenotype with mutations of cholecystokinin-1 receptor were used as a model of T2DM [14]. Male LETO and OLETF rats were housed two per cage in 20 ± 2.5 °C of a temperature and 12:12 hour light–dark cycle of light room, provided with adequate food and water, and monitored for body weight and food intake. Rats were fed either a standard chow diet (SC; 10 kcal% fat, D12492 diet match 7% sucrose, research diet, USA; OLETF controls) or a standard chow (SC) diet containing curcumin (5 g/kg diet, Sigma, Cat # C1386). OLETF rats were randomly assigned to three groups (OLETF control group; *n* = 10), the diabetic exercised group (OLETF+EX group; *n* = 10), or the diabetic exercised plus curcumin fed group (OLETF+EX+Curcumin group; *n* = 10) at 22 weeks of age. Male LETO rats were matched for age. All animal experiment performance was approved by the Institutional Animal Use and Care Committee in Hannam University (HNU2016-16).

### 2.2. Exercise Protocol

The rats in OLETF+EX group were trained to climb a 1 m vertical (85° inclined) ladder with weights secured to their tails. During the first week, rats were familiarized with climbing up to the top cage with or without a weight. Training sessions, from the second week were started with a weight of 50% of body weight. Rats began climbing from the bottom of the ladder and were forced to climb to the top. When they reached the top, a 2 min rest period was allowed before the next trial. Subsequent trials were performed from the bottom of the ladder and 20 g was added to the previous weight at every trial. If a rat was able to climb 10 times with increasing weights, the training session was considered complete. In the event of failure to climb, the rat was forced to complete 10 trials with the last successful weight with no further weight increases [15].

### 2.3. Morris Water Maze Task

Spatial learning and memory were investigated using the Morris water maze task [16]. A circular tank was in a diameter 136 cm, a height 60 cm, and a depth of 30 cm filled with 24 ± 1 °C water. The maze was divided into four equal quadrants and each quadrant of release points was assigned in four points of north (N), east (E), south (S), and west (W). A hidden circular platform (10 cm in a diameter) submerged 1.5 cm beneath the surface of the water was located in the center of the southeast quadrant. Various locations provided visual cues around the maze. Escape latency and traveled distance were measured for a tracking system. The OLETF and LETO rats performed four trials per day for five consecutive days on the platform located in the same position. Randomly, we assigned the starting points each day. We terminated trials either when OLETF rats and LETO rats climbed onto the platform or when time of 60 s had elapsed. After climbing onto the platform, the OLETF rats and LETO rats were allowed to remain on the platform for 15 s. When a rat did not find the platform within 60 s, the rat was moved to the platform and assigned a score of 60 s. After completing four trials, OLETF and LETO rats returned to their own cages with a warm environment. Escape latencies and traveled distances on the final platform were calculated using a computerized video-tracking system (Noldus Ethovision Version 5, Wageningen, The Netherlands). We defined daily scores as the average scores of four trials performed on the day. OLETF and LETO rats performed a 60 s probe trail with the platform removed from the tank on the data. Time spent in the target quadrant was calculated using a computerized video-tracking system (Noldus Ethovision Version 5, Wageningen, The Netherlands).

### 2.4. Blood Biochemistry

Briefly, the experimental rats were euthanized by isoflurane and collected fasting whole blood samples overnight. The blood of experimental rats was obtained using a 20 gauge through the left lateral approach on cardiac puncture. The serum was obtained by whole blood 9 mL in empty tubes and allowed the blood to clot completely. We centrifuged each tube at 2500× *g* for 15 min and transferred the serum into an empty tube for biochemistry analysis. We analyzed the fasting glucose levels using a One Touch II glucose meter (Life scan, Johnson & Johnson, New Brunswick, NJ, USA). The serum insulin, serum total cholesterol, and triglyceride (TG) levels were obtained on an enzymatically-linked immunoassay (ELISA; ALPCO, Salem, NH, USA) and enzymatic kits (Wako Chemicals USA, Inc., Richmond, VA, USA).

### 2.5. Cell Cycle Analysis

Rat intestines were obtained after sacrifice when rats had been fed for 8 weeks. Intestinal epithelial cells (IEC) were isolated using Dispase (Sigma–Aldrich, St. Louis, Missouri, USA) and a 45 μm cell strainer, washed twice with PBS, fixed with 70% ethanol, and stained with Propidium Iodide staining (PI; Invitrogen, Waltham, Massachusetts, USA) for fluorescence-activated cell sorting (FACS) analysis.

### 2.6. Hematoxylin and Eosin (H&E) Staining

Rat liver, skeletal muscle, small intestines, and skin tissues were fixed in 4% paraformaldehyde and embedded in paraffin blocks. Blocks were sectioned at 3 μm, and sections were stained with hematoxylin and eosin (H&E). Morphological analyses were conducted using the Leica Qwin image analyzer system equipped with a Leica DMLS microscopy.

### 2.7. RNA Extraction and cDNA Synthesis

TRI reagent from Bio-Fact (Daejeon, Korea) was used to isolate total RNA from 30 mg whole intestine tissue according to the manufacturer’s instructions. RNA was quantified with Nano Drop (Nano Drop ONE, Thermo Scientific Inc., Waltham, Massachusetts, USA). An RT-Kit (M-MLV, RNase H-, Bio-Fact Co., Daejeon, Korea) was used for cDNA synthesis.

### 2.8. Quantitative Real-Time qPCR

mRNA expression levels determined by Real-Time PCR (Agilent Co., Santa Clara, CA., USA) using 2X Real-Time PCR Master Mix (Including SYBR Green I, Low ROX, Bio-Fact Co.). Data analysis was performed using the Agilent Aria MX 1.0 program. Primer sequences for rat IL6, TNFα, IL10, BiP and β-actin were as follows; (IL6) fw: TCC TAC CCC AAC TTC CAA TGC TC, rv: TTG GAT GGT CTT GGT CCT TAG CC, (TNFα) fw: GTC GTA GCA AAC CAC CAA G, rv: AGA GAA CCT GGG AGT AGA TAA G, (IL10) fw: ATT GAA CCA CCC GGC ATC TA, rv: CAA CGA GGT TTT CCA AGG AG, and (BiP) fw: AGA AAC TCC GGC GTG AGG TAG A, rv: TTT CTG GAC AGG TTT CAT GGT AG, (β-actin) fw: GGC ACC ACA CTT TCT ACA AT, rv: AGG TCT CAA ACA TGA TCT GG

### 2.9. Reverse Transcription-PCR

Reverse transcription (RT)-PCR was performed using 2X Taq Basic PCR Master Mix according to the manufacturer’s instruction (BioFACT Co., Korea). Samples were loaded into a 3% agarose gel for electrophoresis and visualized under UV light (AE-9000 E-Graph, ATTO, Tokyo, Japan). The primer sequences of rat CHOP and hypoxanthine phosphoribosyltransferase (HPRT) were as follows; (CHOP) fw: GTC TCT GCC TTT CGC CTT TG, rv: AGC TGG AAG CCT GGT ATG AGG A, (HPRT) fw: GCT GAC CTG CTG GAT TAC AT, rv: CCC GTT GAC TGG TCA TTA CA.

### 2.10. Statistical Analysis

Statistical analysis was performed in using one-way ANOVA for all data and using two-way repeated measures ANOVA followed by Tukey’s post-hoc test for behavioral data. Results are presented as means ± standard errors, and statistical significance was approved for above 0.05 of *p*-values

## 3. Results

### 3.1. Effects of Exercise and Curcumin Supplementation on Metabolic Complications in OLETF Rats

We used Otsuka Long-Evans Tokushima Fatty (OLETF) rats, which is a well-established model of Type 2 diabetes mellitus (T2DM) and Long-Evans Tokushima Otsuka (LETO) rats as non-diabetic controls. OLETF rats, generated from the Charles River Laboratory and the Tokushima Research Institute, develop diabetes spontaneously with polyuria, polydipsia, and mild obesity phenotypes [17]. Initially, we examined glucose and lipid metabolisms in OLEFT and LETO rats and then examined whether exercise and curcumin supplementation had beneficial effects on OLETF rats.

We measured changes in body weight, fasting glucose, and HOMA-IR scores (Figure 1A,B). Mean final body weight was significantly higher for OLETF controls than LETO controls (*p* < 0.05). Mean body weight was significantly lower in the OLETF+EX+Curcumin group than in OLETF controls (*p* < 0.05), whereas no significant difference in mean body weights was observed between OLETF controls and rats in the OLETF+EX group (Figure 1A,B). Average of food intake (g/day) was significantly higher in OLETF controls than LETO controls (*p* < 0.05) as expected, and there were no significant differences among OLETF controls, the OLETF+EX, and the OLETF+EX+Curcumin group (Figure 1C). OLETF controls had significantly higher fasting blood glucose and HOMA-IR levels than LETO controls, and fasting blood glucose and HOMA-IR levels were significantly lower in the OLETF+EX and the OLETF+EX+Curcumin groups than in OLETF controls, and were lower in the OLETF+EX+Curcumin group than in the OLETF+EX group (*p* < 0.05; Figure 1D,E).

To determine whether exercise plus curcumin supplementation has a beneficial effect on lipid metabolism in OLETF rats, serum triglyceride and total cholesterol levels were analyzed. OLETF controls had significantly higher triglyceride (*p* < 0.05) and total cholesterol levels (*p* < 0.05) than LETO controls. Interestingly, triglyceride and total cholesterol were significantly lower in the OLETF+EX+Curcumin group than in OLETF controls, but no significant difference was observed between in the OLETF+EX group and OLETF controls (Figure 1F,G).

### 3.2. Effects of Exercise Plus Curcumin Supplementation on Histopathology in OLETF Rats

We examined the effects of exercise plus curcumin supplementation on morphological changes in hepatic and muscle tissues by H&E staining (Figure 2). Hepatic tissues of OLETF controls showed mild vacuolation, whereas this was not observed in LETO controls. Interestingly, in the OLETF+EX+Curcumin group, hepatic and muscle tissues appeared normal without vacuolation (Figure 2A). Neither LETO nor OLETF controls exhibited abnormal characteristics such as muscle fibrosis of gastrocnemius muscles. However, muscle fiber sizes were significantly smaller in OLETF controls compared to LETO controls (*p* < 0.05) and muscle fiber size of the both the OLETF+EX and the OLETF+EX+Curcumin group was significantly bigger in OLETF controls. In skin sections, OLETF controls were found to have less developed skin muscle layers under hypodermis than LETO controls. Interestingly, these skin muscle layers were much more developed in the OLETF+EX and the OLETF+EX+Curcumin groups than in OLETF controls.

T2DM is considered to be accompanied by chronic inflammation, and even mild but chronic inflammation could disrupt small intestine epithelium and shortens villi. Small intestines of OLETF controls showed relatively short and disrupted villi as compared with LETO controls, suggesting that OLETF rats exhibit chronic inflammation in the small intestine. Therefore, this observation showed. In addition, infiltrating monocytes were observed in villi of the small intestine as shown with arrows, which are normally observed in inflammatory conditions such as colitis. 40× magnified photo shows irregularly aligned infiltrated monocytes into the villi in OLETF group. Interestingly, these phenomena were not observed in the OLETF+EX+Curcumin group (Please see in Figure 2A as indicated arrows), whereas the OLETF+EX group showed less intensive monocyte infiltration of villi compared to OLETF group, though villi were of normal length.

### 3.3. Effects of Exercise and Curcumin Supplementation on ER Stress in the Small Intestine Tissues of OLETF Rats

Since the intestine is the first barrier from the foreign substance in the body, and the place where curcumin is absorbed, we decided to examine the small intestine to determine whether curcumin affects the first place in the body regarding diabetic-induced inflammation and ER stress. We examined inflammatory cytokines and ER stress marker levels in the small intestine by region. As shown in Figure 3A, the duodenums of OLETF controls showed up-regulation of proinflammatory cytokines (IL6 and TNFα) and downregulation of anti-inflammatory IL10. Surprisingly, exercise alone did not abolish these effects, but exercise plus curcumin significantly reduced these diabetic effects, suggesting exercise plus curcumin reduced diabetes-associated proinflammatory response and enhanced anti-inflammatory response. Also, the expressions of the ER stress marker BiP (Figure 3A) and the ER stress-induced apoptosis marker, CHOP (Figure 3C) were significantly lower in the OLETF+EX+Curcumin group than in OLETF controls and the OLETF+EX group.

A similar pattern was observed in jejunums. As shown in Figure 3B, IL6 and TNFα levels were significantly higher in OLETF controls than in LETO controls, but this effect was significantly abolished by exercise plus curcumin more than by exercise alone. Also, BiP was significantly lower in the OLETF+EX+Curcumin group than in the OLETF+EX group and in OLETF controls. This data shows that exercise plus curcumin supplementation had a marked effect on diabetes-associated inflammation and ER stress, but that exercise alone did not have this marked effect.

In addition, we examined whether the cell cycle of small intestine cells was different in OLETF and LETO controls. We observed the G0/G1 phase of the cell cycle was attenuated in the OLETF+EX group and OLETF controls compared with LETO controls (Figure 3D), but that the G0/G1 phase was higher in the OLETF+EX+Curcumin group than in the OLETF+EX group and in OLETF controls (*p* < 0.05). These observations suggest that diminished CHOP mRNA levels and BiP mRNA levels induced by exercise plus curcumin supplementation might enhance the G0/G1 cell cycle percentage of cellular survival or proliferation stage, and thereby improve cognitive function in OLETF rats.

### 3.4. Effects of Exercise and Curcumin Supplementation on Learning and Memory Deficits in OLETF Rats

Cognitive function was assessed using the Morris water maze test. Mean escape latency decreased over the consecutive five days of learning period in all groups (Figure 3A). We found that mean escape latency was greater for OLETF than LETO controls (49.3 s versus 11.8 s of OLETF rats versus LETO rats on the tracking system, *p* < 0.05), but significantly smaller in the OLETF+EX+Curcumin group (13.8 s) than in the OLETF+EX group (18.8 s) and in OLETF (49.4 s) controls (Figure 4A).

To evaluate memory retention, we performed a probe trial two days after the last Morris water maze test. Rats were admitted to swim for 60 s in the swim pool in which they trained, but with the escape platform withdrawn. The percentage of swimming time spent within the target quadrant by OLETF controls was smaller than that spent by LETO controls (24.4% versus 48.9% of OLETF rats versus LETO rats on the tracking system, *p* < 0.05), and by rats in the OLETF+EX (40.6%) and the OLETF+EX+Curcumin (45.2%) groups, and was significantly smaller in the OLETF+EX group than in the OLETF+EX+Curcumin group (Figure 4B,C). These results suggest that exercise plus curcumin may improve cognitive function.

## 4. Discussion

In this study, we hypothesized that exercise plus curcumin in combination might synergistically attenuate an inflammatory response in diabetes. Our results present that exercise together with curcumin supplementation attenuated diabetic phenotypes in OLETF rats, as determined by body weight, glucose, insulin, TG, and total cholesterol level measurements. In particular, we found that exercise with curcumin enhanced insulin sensitivity, as determined by HOMA-IR indices, and reduced fasting glucose levels in OLETF rats.

Inflammation is a fundamental biological process that restores tissue homeostasis through various repair mechanisms and influences many acute and chronic pathological conditions. However, proper regulation of inflammatory mechanisms is essential to prevent uncontrolled amplification of initial inflammatory responses, and shift from tissue repair towards collateral damage and disease development [18]. Therefore, mild chronic inflammation is the common phenotype of metabolic syndrome including T2DM. It has been shown that ER stress plays a critical link, as well as common phenomenon, between inflammation and T2DM, and many other metabolic disorders [19]. Although the causes leading to ER stress are still not well defined, it is proposed that the causes are the increases in the circulating proinflammatory cytokines, such as TNFα and IL6 in these metabolic conditions [8]. In T2DM, increased IRE1α signaling activates the NF-kB pathway by phosphorylation of the c-jun N-terminal kinase 1 (JNK), well-known modulators of inflammation by promoting the expression of genes, and the activity of proteins involved in the regulation of inflammatory response. This also results in impaired insulin signaling and activation of several molecules in the skeletal muscle, adipose tissue, and liver [8]. It has been reported that ER stress-impaired insulin signaling leads to systemic insulin resistance, hyperglycemia, and hyperinsulinemia [7]. In the study, our results using OLETF rats, naturally induced T2DM model, confirmed increased insulin resistance represented as HOMA-IR on OLETF controls, and that this phenotype was improved with the combination therapy of exercise and curcumin supplementation.

Diabetes is an example of a pathologic neurodegenerative process associated with chronic inflammatory changes. The underlying mechanisms of diabetics and cognitive deficits are known to involve elevated phosphorylated Tau (p-Tau) protein levels in cerebrospinal fluid and long noncoding RNA-induced apoptosis of hippocampal neurons in older diabetic animals and patients [20,21]. Several studies have shown that exercise reduces the inflammatory response in diabetes brain [22,23,24,25,26]. However, other human studies have shown that single lifestyle factor changes, such as physical activity changes, have less effect on cognitive function in neurodegenerative diseases. Curcumin is a well-known natural antioxidant that reduces the inflammatory response in diabetes via Nrf2 and PI3K/AKT signaling pathways [27,28,29,30,31]. Evidence indicates that curcumin can ameliorate pathologies associated to amyloid-β and recover p-Tau-related pathologies in animal models [32,33]. Also, curcumin has been shown to reduce the gene expression of insulin receptor and insulin receptor substrate-1 in the hippocampus [34,35,36]. Consistently, our results indicate that the combination of exercise and curcumin synergistically improves cognitive function in OLETF rats or inflammation response in OLETF intestines.

In a brain-specific reduced ER stress mouse model, severe leptin resistance significantly increased obesity on a high-fat diet, which suggested that ER stress might participate in the adjustment of insulin sensitivity in brain [37]. With many previous studies, we hypothesized that ER stress observed in the intestine in our study also occurred in the brain, as ER stress occurs in various tissues systemically, and postulated that inhibiting ER stress of the intestine by exercise and curcumin supplementation would happen in the brain tissue, suggesting that indirect inhibiting ER stress in the brain might improve the cognitive function of the brain [18,19,38,39,40,41].

Emerging preclinical evidence has shown that the bidirectional signaling between the gastrointestinal (GI) tract and the brain, the so-called gut–brain axis, plays an important role in both host metabolism and behavior [42]. Evidence suggests that microbial diversity is reduced in metabolic dysregulation [43,44,45]. The microbial metabolites short chain fatty acids are found to exert numerous physiologic effects, including energy homeostasis through the regulation of GI hormones such as cholecystokinin, glucagon-like peptide 1 (GLP1), and leptin [46]. Moreover, intestinal microbiomes cause changes in the intestinal hormone secretion and immune response as well as the intestinal mucous membrane barrier, blood-brain barrier control, and the creation and decomposition of neurotransmitter, thus, closely interacting with the central nervous system, which affects the brain’s ability to develop cognitive behavior. Correlations between intestinal microbiome and brain so-called brain–gut axis have been intensively studied in neurological diseases. One study showed that the loss of toll-like receptor (TLR) 2, one of the sensor for innate immunity, in conventionalized mice results in a reminiscent phenotype of metabolic syndrome, characterized by a clear difference in the gut microbiota, which induces insulin resistance and inflammation associated with ER stress, reproduced in wild-type mice by microbiota transplantation and can be reversed using antibiotics [47]. This data emphasized the role of microbiota in the complex network of molecular and cellular interactions that bridge genotype to phenotype, and have potential implications for a wide array of common human disorders involving diabetes, and even other immunological disorders [43].

However, this study had several limitations. First, the study focused on effect of exercise of OLETF rats for the cognitive function, which does not provide the experimental group of OLETF rats with curcumin alone. Nevertheless, exercise is a major factor of lifestyle for adults in cognitive function, which also included to investigate the effect of curcumin in the study. Second, we didn’t provide the data of histology or ER stress in brain. However, our lab previously reported the OLEF rats with exercise in the hypothalamus show significant increases in mRNA levels of GLP-1 (a major incretin hormone), glucose transporter 2 (GLUT2), and superoxide dismutase 1 (SOD1, antioxidant enzyme) compared to OLETF rats in brain tissues [48]. Indeed, as mentioned above, we found that exercise plus curcumin abolished ER stress response in intestinal tissues of OLETF controls and increased insulin sensitivity in blood system. In addition, rats in the OLETF+EX+Curcumin group exhibited better learning skills and brain memory in the Morris water maze test, which suggests exercise plus curcumin regulates insulin sensitivity in whole body by reducing ER stress and sensitizing leptin resistance. As a result, insulin level is regulated, high-blood glucose-induced ER stress is ameliorated, and ER stress-induced apoptosis is inhibited in blood system and cognition is improved in OLETF rats. Further studies are needed to confirm the results in the various tissues, including the brain, in terms of inflammation and ER stress in order to reduce the symptom of the diabetic disease.

For the third limitation, curcumin, as a bioactive compound, is soluble in ethanol, alkali, acetic acid, and other organ solvents, and is poorly absorbed in the intestine. One study reported absorption of curcumin 400 mg given orally to rats, was detected in 5 mg/L in the hepatic portal vein to 24 h after oral injection [49,50]. However, in this study, we observed that rats fed curcumin with the amount of a 5 g per diet (kg), improves the weight loss, glucose homeostasis and lipid profiles. Despite these limitations, the available data leads to the conclusion that the combination of exercise and curcumin improves diabetics-related cognitive phenomenon in rats.

## 5. Conclusions

We found OLETF rats in combination of exercise with curcumin supplementation improved spontaneously developed Type 2 diabetic characteristics such as enhanced fasting glucose levels, insulin resistance, and lipid profiles. In addition, learning and memory retention deficiencies in OLETF rats were improved by exercise plus curcumin supplementation, presumably by suppressing inflammatory response and ER stress response. We believe these results have potential applications for the nutritional therapy of diabetes-associated cognitive dysfunction.

## Figures and Tables

**Figure 1 nutrients-12-01309-f001:**
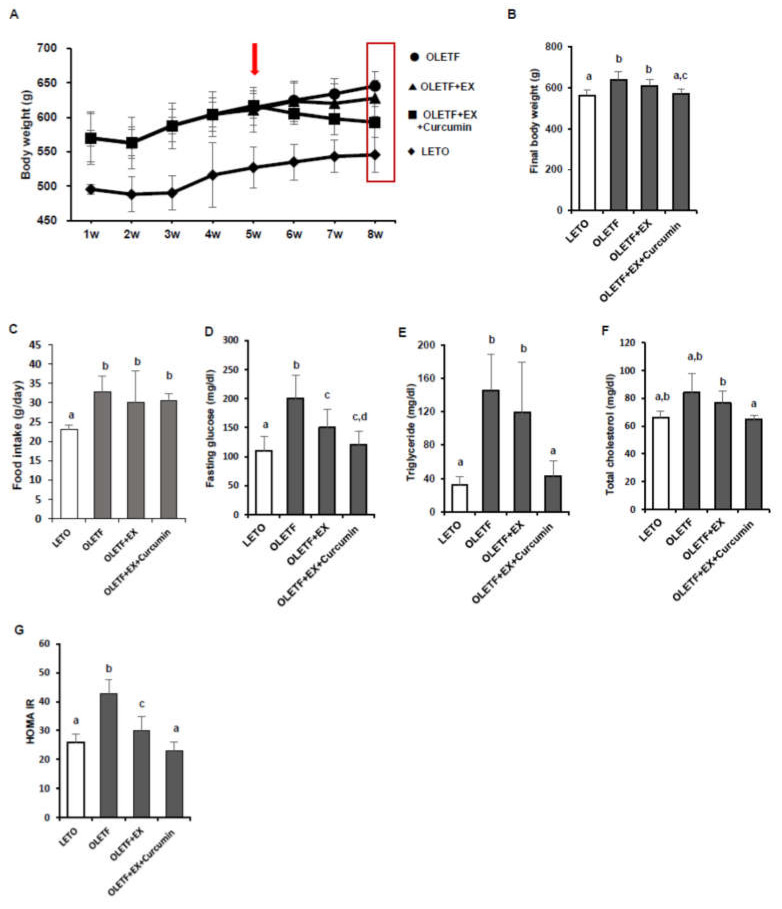
Exercise plus curcumin supplementation reduced body weights and improved glucose homeostasis and lipid profiles in OLETF rats. (**A**) Plot of body weights (g) over the eight weeks experimental for OLETF and LETO controls (10 male rats/group). Arrow indicates a start period of the intervention study in exercise with curcumin supplementation. (**B**) Mean body weight was significantly lower in the OLETF+EX+Curcumin group than for OLETF controls and lower in the OLETF+EX+Curcumin group than in the OLETF+EX group. (**C**) Food intake (g/day) was measured in twice a week. (**D**) Fasting glucose levels were lower in the OLETF+EX+Curcumin and the OLETF+EX groups than in OLETF controls, but no significant difference was observed between these intervention groups. (**E**–**G**) Serum triglyceride, total cholesterol, and Homeostatic Model Assessment of Insulin Resistance (HOMA-IR) in OLETF and LETO controls. Results are presented as means ± Standard deviations (SDs). Significant differences were determined using One-way ANOVA. Different superscripts indicate significant difference; a,b,c,d (*p* < 0.05). LETO: age-matched control group, OLETF: diabetic group, OLETF+EX: diabetic exercise-trained (to climb a ladder with added weights) group, OLETF+EX+Curcumin: diabetic exercise-trained and fed curcumin (5 g/kg/diet) group.

**Figure 2 nutrients-12-01309-f002:**
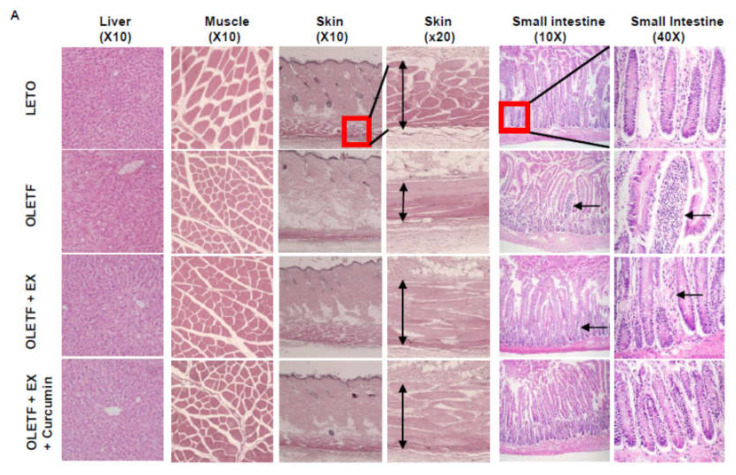
Curcumin reduced diabetic-related phenotypes of liver, muscle, skin and small intestine in diabetes rats. (**A**) Representative hematoxylin and eosin (H&E) staining of liver, muscle, skin, and small intestine tissues of OLETF and LETO controls obtained at the end of the eight weeks experimental period. Exercise plus curcumin changed the morphologies of liver, muscle layers in skin (arrowed) and villi in small intestine (arrowed). (**B**) The quantitate data of muscle fiber size in experimental rats. The cross-sectional area of the muscle fiber was measured for all muscle fibers of five images selected from each animal. In data, the average area of muscle fibers was calculated and presented as the mean area value of total muscle fibers in five images. Significant differences were determined using one-way ANOVA followed by Turkey’s post-hoc test. Different superscripts indicate significant difference (*p* < 0.05). LETO controls: age-matched control group, OLETF controls: diabetic group, OLETF+EX: diabetic exercise-trained group, OLETF+EX+Curcumin: diabetic exercise-trained plus curcumin (5 g/kg in diet).

**Figure 3 nutrients-12-01309-f003:**
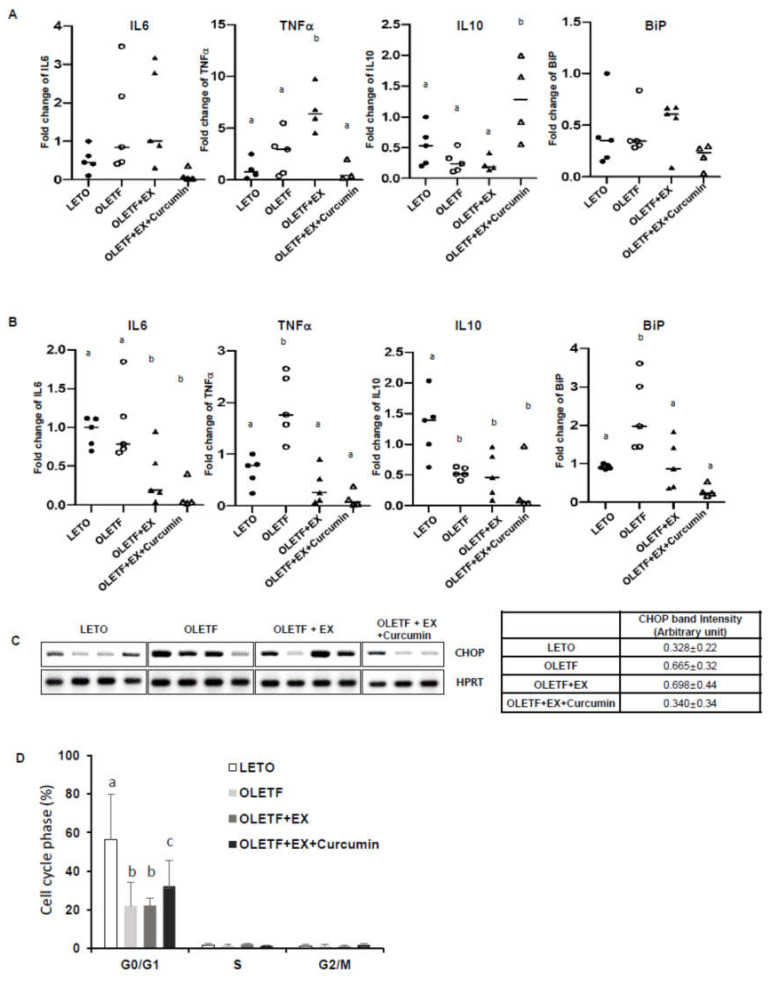
Curcumin supplementation with exercise inhibited proinflammatory cytokines and ER stress in small intestines. (**A**–**C**) The whole duoden ums (**A**,**C**) and jejunums (**B**) were harvested, RNAs were extracted, and cDNAs were synthesized for RT-PCR targeting apoptosis marker C/EBP homology protein (CHOP) (**C**) and for Real Time PCR targeting proinflammatory cytokines (IL6 and TNFα (pro-inflammatory cytokines), IL10 (anti-inflammatory cytokine)), and the ER stress marker BiP (**A**,**B**). Hypoxanthine phosphoribosyltransferase (HPRT) and β-actin were used as loading controls for RT-PCR and Real Time PCR, respectively. (**D**) In OLETF controls, curcumin plus exercise rescued the G0/G1 phase of the cell cycle in isolated small intestine cells. Isolated small intestine cells were fixed with 70% ethanol and stained with Propidium Iodide (PI). Flow cytometry was used to quantify cell phases (%, of Y-axis). Significant differences were determined using a one-way ANOVA followed by Turkey’s post-hoc test. Different superscripts indicate significant difference (*p* < 0.05). Results are presented as means ± standard errors. LETO controls: age-matched control group, OLETF controls: diabetic group, OLETF+EX group: diabetic exercise-trained group, OLETF+EX+Curcumin: diabetic exercise-trained group fed curcumin (5 g/kg in diet).

**Figure 4 nutrients-12-01309-f004:**
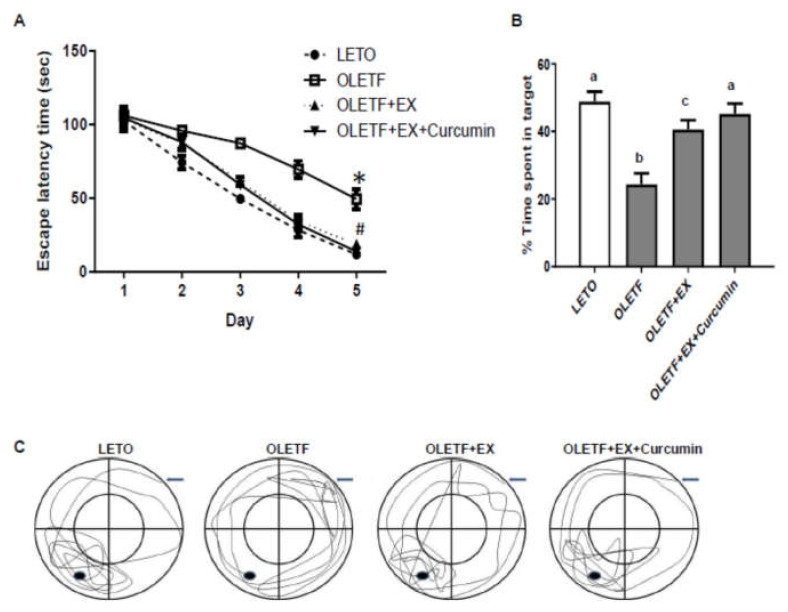
Curcumin plus exercise improved learning and memory retention of OLETF rats in the Morris water maze test. (**A**) Escape latency was measured with the platform in place. In four trials per day, experimental rats performed to obtain data for five consecutive days. * *p* < 0.05 vs. LETO rats of controls, # *p* < 0.05 vs. OLETF rats of controls. (**B**–**C**) The effect of exercise plus curcumin on memory retention in the probe trial. Time spent in the target quadrant in the probe trial was calculated on the sixth day (**B**). The representative diagram of tracking on swim path length. The data were determined with a video tracking system (**C**) LETO controls: age-matched control group, OLETF controls: diabetic group, OLETF+EX group: diabetic exercise-trained group, OLETF+EX+Curcumin group: diabetic exercise-trained and curcumin fed (5 g/kg diet) group. Results are presented as means ± standard errors. Statistical analysis was conducted using one-way ANOVA for the probe trial and by using a two-way repeated measures ANOVA followed by Turkey’s post-hoc test for behavioral data. Different superscripts indicate significant difference (*p* < 0.05).

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
