# Peer review of "Exercise and Curcumin in Combination Improves Cognitive Function and Attenuates ER Stress in Diabetic Rats"

_nutrients, 2020, doi:10.3390/nu12051309_

Round 1

Reviewer 1 Report

In this manuscript, the authors indicated the new approach against cognitive function in diabetic model rats using curcumin. They also showed that the combination of exercise and curcumin supplementation attenuates ER stress in the intestine.

Major comments;

  1. The authors haven’t shown the results of “curcumin only” OLETF rats. Please add the results or description if the experiments have been done, or please mention the reason why the experiments haven’t done.

  1. The authors showed the attenuated ER stress and activated cell cycle just in intestine samples.
  • To state “thereby improve cognitive function in OLETF rat” as written on page 8 line 283, authors need to show any change in histology or ER stress in brain tissue.
  • The authors need to show the curcumin effect on the pancreas. How did the curcumin supplementation change the beta-cell mass, beta-cell proliferation, and ER stress in beta-cells?

Minor comments;

  1. I assume that the absorption rate of curcumin from the intestinal tract is poor. How much curcumin did the mice take?

  1. The authors need to show changes in the mice's food intake (curcumin might have a distinctive taste).

  1. Please add the description of the fasting time duration and how the serum collection was performed.

  1. In Fig 1A, the error bar needs to be added in each line graph.

  1. In Fig 2, please put the magnifications by the tissue types.

  1. In Fig 3 legend, what does “a,b,c,d; Different letters indicate significant intergroup differences” mean?

  1. On page 6 line 232 and line 239, what does “F (12,144) = 35.080 of coordinate system” and “F (3,39) =131.450 of coordinate system mean” mean?

  1. On page 6 line 230, the sentence “We observed that changes in latencies revealed a significant time among group (p=.000)” doesn’t make sense.

  1. In Fig 4C, CHOP expression needs to be quantified.

  1. In Fig 4D, there is no error bar on the “OLETF+EX+Curcumin” group bar graphs. Does it mean it was just one sample for this group? If so, how did the authors calculated the statistical difference?

  1. Throughout the manuscript, several prepositions are left out in several places.

Author Response

Dear reviewer

I wish to thank you and the reviewers for a favorable review of our manuscript entitled, “ Exercise and curcumin in combination improves cognitive function and attenuates ER stress in diabetic rats”.

We have improved the manuscript to meet the reviewer’ criticisms.

Importantly, we have provided new experimental data addressing the reviewer’ concerns. Specifically, we have provided these new figures:

Figure 1 C provides the average of food intake of experimental groups.

We will discuss these new figures within the context of the point-by-point response to the reviewers, which is attached to this letter.

Thank you again for these thorough reviews. We hope the work is now suitable for publication in Nutrients.

Sincerely yours,

Eunmi Park, Ph.D.

Reviewer 2 Report

I have read the manuscript of Cho et al with interest. Curcumin attracts a lot of attention for its potential beneficial properties for treatment of several human diseases that are, however, challenging to pinpoint. Therefore, animal studies exploring that are important and interesting to the audience. The article is well-written, and the research is diligent. Please address my specific points below:

1) The main question I had regarding this manuscript is that the authors failed to establish connection between gut effects of curcumin and improved brain function. All histopathology and biochemistry was done in the intestine (and skin). In the Results, please add the rationale for choosing these particular tissues for analysis. If no inflammatory markers could be studied in the brain, the authors should at least address this question in the Discussion part. What is the gut-brain connection? What recent studies supporting it have been published? Is there a possible role for gut microflora?

2) Results lines 161-163: please remove the template text;

3) Line 256: I would suggest changing the order of chapters 3.3 and 3.4, putting the cognitive test at the end, where it would look more appropriate.

4) Chapter 3.2, Figure 2: The monocyte infiltration is not clearly seen on the Figure due to low resolution, and the conclusions seem to be too strong. Muscle fibers size said to be "significantly" smaller in OLETF rats, while no quantification/measurement is presented. Please consider improving the figure quality and correcting the text.

5) Chapter 3.4, Figure 4: (A, B) - for each graph, please show the cytokine name in the title, otherwise it is not clearly visible; (C) - it is not clear why CHOP expression is presented differently. What are the 4 and 3 bands presented for each of the 4 conditions? Is it possible to quantify the bands (e.g. line reader software for gels) and present in a graph rather than on the image? (D) Color-coding does not work. Please use pattern coding for the 4 conditions instead. Please add the error bars for the OLETF+Ex+Curcumin condition or explain why they are not presented. In is not clear, where the "*" (p-value) in the footnote belongs to on the graph. It is not clear, what the letters "a,b,c" on the graph refer to neither. Please improve this figure, it should be self-explanatory.

6) Discussion: repeating the point 1), please present the gut-brain connection (incl. through inflammation) in more detail, citing the recent literature on the subject.

Author Response

Dear reviewer

I wish to thank the reviewer for a favorable review of our manuscript entitled, “ Exercise and curcumin in combination improves cognitive function and attenuates ER stress in diabetic rats”.

We have improved the manuscript to meet the reviewer’ criticisms.

Importantly, we have provided new experimental data addressing the reviewer’ concerns. Specifically, we have provided these new figures:

Figure 2 A and B provides the new photo of intestine H&E staining and the quantification of muscle fiber size.

Figure 3 C provides the quantification of CHOP band.

We will discuss these new figures within the context of the point-by-point response to the reviewer, which is attached to this letter.

Thank you again for these thorough reviews. We hope the work is now suitable for publication in Nutrients.

Sincerely yours,

Eunmi Park, Ph.D.

Round 2

Reviewer 1 Report

The authors added some figures and descriptions of the limitation of this research design. The raised points had all adequately assessed by the authors.
Two minor corrections might be needed.
1. Page2 line 85. three "group groups"
2. Page3 line 111. LE"OT" rats (I think this should be LE"TO")

Author Response

Dear Reviewer #1

We appreciated your valuable review again.

We fixed two minor corrections as requested ;

Two minor corrections might be needed.

1. Page2 line 85. three "group groups"

Yes, we fixed words for “ three groups” on line 85 in the revised manuscript.

1.Page3 line 111. LE"OT" rats (I think this should be LE"TO")

Yes, we changed  for “ LETO” on line 111 in the revised manuscript.